# The Impact of Accessibility of Community Elderly Care Services on Quality of Life of the Elderly

**DOI:** 10.3390/healthcare13020099

**Published:** 2025-01-07

**Authors:** Xiaodong Di, Lijian Wang

**Affiliations:** School of Public Policy and Administration, Xi’an Jiaotong University, No 28 Xianning West Road, Xi’an 710049, China; wanglijian2@mail.xjtu.edu.cn

**Keywords:** accessibility, community elderly care services, service utilization, quality of life of the elderly, influence mechanism

## Abstract

**Background/Objectives**: With the gradual increase in population aging and the prevalence of the empty nest, community elderly care services have become an effective service model for responding to population aging because they can alleviate the decline in family care function and meet the needs of elderly homecare patients. This paper aims to identify the influence mechanism of the accessibility of community elderly care services on the quality of life of the elderly. **Methods:** This paper surveyed a total of 949 elderly people and adopted the interview questionnaire survey method, and it used ordered logistic regression to analyze the impact of accessibility on the quality of life of the elderly. **Results:** The study finds that approachability, availability, acceptability, and accommodation affect the living environment satisfaction of the elderly. Accessibility affects environmental satisfaction by influencing the service utilization frequency, and accessibility affects the psychological health and environmental satisfaction of the elderly by influencing service utilization satisfaction. **Conclusions:** The improvement of accessibility can increase service utilization and, thus, improve the welfare of the elderly. So, the government should improve the accessibility of community elderly care services.

## 1. Introduction

Population aging is a major feature of the demographic changes in the world and an important factor in achieving high-quality economic development. In China, by the end of 2022, there were 280.04 million people over the age of 60, accounting for 19.8 percent of China’s population, and 209.78 million people over the age of 65, accounting for 14.9 percent. By 2025, there will be 300 million people over the age of 60, accounting for 21 percent, and 15 percent for people over the age of 65 [1]. With the gradual deepening of population aging and the empty nest, the Chinese government has attached more importance to the construction and development of elderly care services. Community elderly care services have become an effective service model for coping with population aging because they can alleviate the decline in family care function and meet the needs of elderly home care [2,3]. However, at present, the development of community elderly care service faces practical problems such as an insufficient supply of elderly care service resources, homogeneity of elderly care service contents, a low utilization rate of elderly care service facilities, and an imbalance of elderly care services between urban and rural areas [4,5]. These problems severely reduce the quality of life of the elderly. Therefore, from the perspective of accessibility, exploring the welfare effect of community elderly care services has become the theme of healthy aging. Only by improving the accessibility of community elderly care services can the development of elderly care services be promoted effectively.

However, the current construction of the accessibility of community elderly care services mainly concentrates on community elderly fitness equipment, home care beds, home care intelligent equipment, and other elderly service facilities. Community elderly care services emphasize the coverage of facilities but ignore the accessibility, such as the acceptability of elderly care service contents, the operability of products for the elderly, the quality of elderly care workers, and the medical care for the elderly, which has led to an obvious imbalance between supply and demand and low efficiency in the utilization of services [6,7], for example, low participation in rural elderly care centers, tedious reimbursement standards and procedures for medical care for the elderly, the low signing rate of family doctors, the low acceptance of long-term care insurance, and so on [8]. Therefore, how to improve the accessibility of community elderly care services becomes a key issue to improve the quality of life of the elderly.

In order to elucidate the influence mechanism of the accessibility of community elderly care services on the quality of life of the elderly, firstly, this paper clarifies the analysis dimension of the accessibility of community elderly care services and puts forward a theoretical hypothesis by analyzing the welfare effect of community elderly care services. Then, it empirically analyzes the impact of different dimensions of accessibility on the quality of life of the elderly. Finally, based on the empirical results, this paper discusses how to improve the welfare of the elderly from the aspects of high-quality development, access construction, and efficient utilization of community elderly care services.

## 2. Theoretical Analysis and Research Hypothesis

### 2.1. Analysis Dimension of Accessibility

The topic of accessibility first appeared in the field of public health, but there is no explicit definition of accessibility. As the research field of accessibility expands, the concept of accessibility becomes more and more specific. In 1981, Penchansky R and Thomas J defined accessibility as the degree of fit between the user and the service system. Accessibility reflects the extent to which a service is used by the target group. Accessibility contains two meanings: One is whether the service is provided, and the other is whether the user can access and use the service when it exists. When the service is not provided, the service is inaccessible, and though the service exists, the service is also inaccessible when the user has obstacles that prevent users from accessing and using the service in the process of using the service. Therefore, whether the service is accessible is whether the user can obtain the service successfully, which is affected by the resource configuration form, the service content, the service cost, the service effect, and so on. Penchansky R and Thomas J generalized these factors into five dimensions: approachability, availability, acceptability, accommodation, and affordability [9].

As typical quasi-public goods, community elderly care services have competitive attributes and public attributes. The accessibility of community elderly care services in this paper mainly refers to the accessibility of community elderly care service resources with public attributes. Approachability of community elderly care services refers to the convenience of the elderly approaching and using elderly care service resources, so specific indicators of accessibility include the travel time, distance, and cost between elderly care service resources and the elderly people living at home. Availability refers to the adequacy of elderly care service resources, including elderly care facilities, elderly care nurses, elderly care service information platforms, and other resources. Acceptability refers to the acceptability of the elderly to the attributes and characteristics of elderly care service resources, including the acceptance of the attributes of service facilities, the service personnel, and the service contents. Accommodation refers to the accommodation of the elderly to the elderly care form and process, including appointments, telephone services, service standards, service supply forms, and so on. Affordability refers to the ability of the elderly to bear the prices and expenses of elderly care services, including the medical price, family income, health insurance, and other costs [10].

### 2.2. Analysis Framework

The capability approach was first proposed by Amartya Sen in 1980. It holds that happiness is not the external material condition but the independent ability to use resources and enjoy services. It emphasizes that quality of life depends on the individual’s ability to do something [11]. According to the capability approach, the capability of the elderly to participate in elderly care services refers to all the abilities of the elderly to reach, select, and participate in elderly care services. The quality of life of the elderly not only depends on the ability to live independently but also on the ability to provide resources for the environment [12]. The needs of the elderly for the environment include physical needs, psychological comfort, social relationships, transportation, and other aspects, which can maintain daily life and social activities and meet social values, privacy, freedom, and other high-level pursuits. Therefore, the influence mechanism of accessibility on the quality of life of the elderly can be analyzed from four aspects: physical control ability, psychological adjustment ability, social participation ability, and environmental adaptation ability.

From the perspective of the physical control ability of the elderly, the elderly are seriously bound by their living background, educational experience, family environment, etc., so their ability to accept and adapt to new things is relatively low. Most of them prefer to choose social activities that are easy to operate and understand, which reflects the capability of the elderly to participate in innovative, intelligent, high-tech, and other activities as limited. In addition, the elderly have weak self-care abilities and need more daily care and support from others. Especially for device-helping or disabled elderly people, their capability to meet daily needs is difficult, which not only needs long-term care from family members but also greatly increases the family’s medical expenses and care costs [13]. Therefore, the improvement of the accessibility of community elderly care services can effectively enhance personal cognitive ability, choice ability, decision-making ability, accommodation, and other feasible abilities of the elderly, which directly affects the quality of life of the elderly. Seangpraw and Ong-Artborirak, based on the Andersen model, found that community elderly care services could not only directly alleviate the health vulnerability of the elderly but could also reduce the financial pressure on the elderly [14]. Van et al. based on the methodological quality of intervention studies, found that community elderly care services could significantly improve the physical health of the elderly and found that daily care and spiritual comfort had a more obvious effect on the health improvement of the elderly [15]. So, we can come up with hypothesis 1 that the accessibility of community elderly care services can improve the physical health of the elderly.

From the perspective of the psychological adjustment ability of the elderly, according to the literature, negative events beyond the individual’s control are strongly related to the strain index, and emotional control can improve the elderly’s sleep and happiness [16]. As the elderly are more prone to a lack of social, economic, and interpersonal resources, personal emotions are easily affected by environmental changes, including physical vulnerability caused by aging, non-accidental events experienced by the elderly, limited opportunities to exercise control, and negative social stereotypes of the elderly [17]. Therefore, a pessimistic mentality like depression, emotional fluctuation, loneliness, and so on, caused by lacking psychological adjustment ability, greatly reduces the quality of life of the elderly. The improvement of the accessibility of community elderly care services can help the elderly improve their psychological recovery ability by studying the dynamic components of the transaction between the elderly and the environment. Esmaeilzadeh and Oz adopted the generalized structural equation method and found that the accessibility of community elderly care services significantly reduced anxiety levels and improved the mental health of the elderly who did not have care needs at present [18]. Boström et al. adopted the ordered logistic regressions method and found that community elderly care support could improve the mental health of the elderly by protecting their cognitive function and promoting social communication [19]. So, we can come up with hypothesis 2 that the accessibility of community elderly care services can improve the psychological health of the elderly.

From the perspective of the social participation ability of the elderly, as an independent individual, the elderly need to rely on others in daily life, but they have their own lifestyle, habits, hobbies, and social contacts and need a certain self-space and freedom. For healthy elderly people, their action capacity can guarantee their autonomy in daily life, but disabled and device-helping elderly people need more respect and understanding from others. In the process of daily care and care service, they should pay more attention to their social needs and self-esteem needs [20]. In addition, the elderly with rich life and work experience are more willing to participate in social activities, show their social status and social role, and assume certain social responsibilities, such as mutual support for the elderly, participation in community management, organizing activities for the elderly, and voluntary activities, which can expand the social network of the elderly. These activities not only reduce the pressure of family care but also meet the needs of the elderly to pursue social values. Jiang and Liu pointed out that the social participation issues of the elderly were largely caused by social–ecological factors, which were mainly constrained by such factors as a lack of public space, insufficient resources for the elderly in the community, and single content of the care service [21]. Zhong and Cheng based on the MIMIC model by groups, found that community social capital was positively correlated with the life satisfaction of the elderly, and participation in social organization activities can help the elderly regain their social identity and sense of value [22]. So, we can come up with hypothesis 3 that the accessibility of community elderly care services can improve the social relationships of the elderly.

From the perspective of the environmental adaptation ability of the elderly, the increase in public facilities for elderly care services, the elderly-oriented environmental reform, and the enrichment of community elderly care service content can greatly improve the care facility and community environment. A good community environment can effectively increase the social participation and neighborhood interaction of the elderly. Convenient transportation and sufficient elderly care resources can accelerate the circulation of elderly care service information, enrich the supply of elderly care service contents, and increase the selection range of elderly care service models, then improve the freedom of the elderly. Efficient service processes and high-quality elderly care services can improve the trust and dependence of the elderly on community services and increase the cohesion between residents and the community, as well as between residents [23]. Therefore, the improvement of the accessibility of elderly care services can release family pressure, replace the family care function with community care, and improve the life satisfaction of the elderly. Kiik and Nuwa based on a cross-sectional study, found that the level of community elderly care service facilities could significantly improve the subjective welfare of the elderly, and the welfare effects of different types of community elderly care service facilities were different [24]. Gu et al. found that community elderly care services could effectively improve the elderly’s environmental satisfaction, and the more service items, the higher the elderly’s satisfaction [25]. So, we can come up with hypothesis 4 that the accessibility of community elderly care services can improve the environmental satisfaction of the elderly.

## 3. Materials and Methods

### 3.1. Data Source

In this article, the data come from the social survey of the major project of the Ministry of Education of China in July 2019. The survey team consists of 25 people, including teachers, doctoral students, and master’s students. The survey members have received professional training and participated in the pre-survey. The survey adopts stratified sampling method and investigates three representative areas of Hanzhong, Baoji, and Yan’an in Shaanxi province, which are characterized by high aging and rapid development of elderly care services. According to the data of the China Statistical Yearbook in 2023, the elderly dependency ratio in Shaanxi Province is 21.42, which is basically the same as the Chinese level of 21.83. Therefore, taking Shaanxi Province as an example to analyze the development status of China’s community elderly care service has a certain representative.

The questionnaire is specially designed according to the content of the accessibility of elderly care services. The contents of the questionnaire are mainly divided into three modules. The first part is the basic information about the elderly, including age, gender, health status, economic income, and family status. The second part is the demand and supply of elderly care services, which mainly includes the construction and utilization of elderly care services such as daily care, medical care, leisure and entertainment, spiritual comfort, elderly rights, and so on. The third part is the satisfaction evaluation of elderly care services, which mainly includes the evaluation of the elderly on the service subjects, service process, service facilities, service personnel, service supervision, and fairness perception. In the process of survey, we adopted the method of random interviews. Each interview group had two members, one responsible for interviewing the elderly and the other responsible for filling in the questionnaire. After the interview, interview group cross-checked the filling results of the questionnaire to ensure the accuracy of the questionnaire.

### 3.2. Variable Selection

Dependent variable. This paper takes the quality of life of the elderly as the dependent variable. Based on the definition of the quality of life of the elderly by the World Health Organization, this paper divides the quality of life of the elderly into four aspects: physical condition, psychological status, social relationship, and living environmental satisfaction [26]. The physical condition of the elderly directly determines the elderly’s mobility and life autonomy. Psychological status of the elderly reflects the satisfaction of the elderly with their self-cognition and social status. Social relationship of the elderly people with neighborhood, kinship, and so on can reflect their sense of social integration and social gain. Living environmental satisfaction of the elderly can reflect the construction situation and service quality of community elderly care services.

Independent variable. This paper takes the accessibility of community elderly care services as the dependent variable, including five dimensions of approachability, availability, acceptability, accommodation, and affordability [27].

Control variables. Considering that the individual characteristics of the elderly have a great impact on the quality of life of the elderly, we choose age, gender, education level, and marital status as control variables to ensure that the results are not affected by the personal characteristics of the elderly.

Regarding the above variables, we use the Likert method to evaluate indicators, and we divide the options into five levels from weak to strong.

### 3.3. Research Method

Firstly, in order to verify the direct welfare effect of the accessibility of community elderly care services, this paper, respectively, takes physical condition, psychological status, social relationship, and living environmental satisfaction as dependent variables and describes the impact of each dimension on the quality of life of the elderly by constructing an ordered logistic regression equation [28]. Ordered logistic regression is a Logit model based on cumulative distribution. Assuming that the dependent variable is an ordered value assigned from 1 to *J*, the cumulative Logit of the dependent variable $y_{i}\leq j\left|x_{i}\right.$ and $y_{i}>j\left|x_{i}\right.$ can be expressed as its basic theoretical model (1).
(1)$$l_{j}(x_{j})=\log\left[\frac{\mathrm{Pr}(y_{i}\leq j\left|x_{i})\right.}{\mathrm{Pr}(y_{i}>j\left|x_{i})\right.}\right]=\alpha_{j}+\beta X$$

In Formula (1), X denotes independent variables, specifically including core independent variables and control variables. β denotes the matrix of coefficients corresponding to X. j denotes the sequence number assigned to the dependent variable from 1 to J. α denotes the intercept term.

According to the basic theoretical model, we can construct the following regression Equation (2).
(2)$$Y=\alpha_{1}ACC+\alpha_{2}X+\varepsilon$$

In the equation, Y denotes the dependent variable, that is, the welfare of the elderly and its sub-dimensions, ACC denotes the independent variable, that is, the five dimensions of accessibility, X denotes the control variable, α denotes the regression coefficient, and ε denotes the error term.

Then, this paper further explores the influence mechanism of accessibility on the quality of life of the elderly; from the perspective of service utilization, we take the service utilization frequency and service utilization satisfaction as the mediator variable to analyze the welfare effect of the accessibility of community elderly care services. So, we can construct the following regression equation.
(3)$$Y=\beta_{1}SU+\beta_{2}X+\varepsilon_{1}$$


(4)
$$SU=\phi_{1}ACC+\phi_{2}X+\varepsilon_{2}$$



(5)
$$Y=\lambda_{1}SU+\lambda_{2}ACC+\lambda_{3}X+\varepsilon_{3}$$


In the equation, SU denotes service utilization, ACC denotes accessibility, X denotes control variable, β, ϕ, λ denote regression coefficients, and ε denotes error term.

Finally, this paper used stata15 software to analyze direct impact and indirect impact.

## 4. Results and Discussion

### 4.1. Characteristics of the Sample

We surveyed a total of 949 elderly people and adopted the interview questionnaire survey method, so a total of 949 questionnaires were collected. However, due to the sampling of residents with different backgrounds and different education levels, some contents of the survey data had problems such as missing answers, omissions, and misfilling. Considering the accuracy of the data analysis, 289 pieces of survey data with obviously missing data were excluded, and other survey data with a small amount of missing data were all included in the analysis model and supplemented according to known information. Maximum statistical integrity is guaranteed. The Cronbach’s α value of the final sample data is 0.68, and the basic information of the sample is shown in Table 1.

From the perspective of variable characteristics in Table 2, the standard deviation of most variables is close to 1, indicating that the data are concentrated near a certain central value, and the sample has no outliers. In terms of accessibility, the mean value and median value of availability are the lowest, 3.05 and 3, respectively, which shows that the infrastructure and products of community elderly care services are relatively lacking. From the perspective of the quality of life of the elderly, the mean value of environment satisfaction, in general, indicates that the elderly have low satisfaction with resources such as community environments, public resources, and interactive places. On the whole, the sample conforms to the actual situation of older adults and has a certain reliability.

### 4.2. Direct Impact

According to the results in Table 3, with the control variables of age, gender, marriage, and education gradually added, the Log-likelihood value of each model is significant, and most of the major variables in the model pass the significance test level of 5%, indicating that the model is reasonable and effective, and the results have a certain stability.

The empirical results show that the acceptability, affordability, and availability of community elderly care services have a significant impact on the physical condition of the elderly; that is, for every 1% increase in the approachability, affordability, and availability of community elderly care services, the incidence rate of the elderly’s physical health improves by at least one level increased by 14.91%, 28.20%, and 15.93%. The result reflects that the resource attributes, service costs, and number of facilities of elderly care services affect the health of the elderly, indicating that the adequacy and operability of elderly care service facilities will affect the elderly’s demand and satisfaction in participating in fitness activities. Approachability and accommodation have no significant impact on the physical condition of the elderly, perhaps because the convenience of the elderly service center and the form of the elderly care service have little impact on the perception of collective social capital of the elderly, and the type, quantity, and cost of elderly care service resources are the key factors that affect whether the elderly are willing to participate in exercise [29]. In addition, the application of smart technologies, such as telemedicine or smart home technology, can exert the subjective initiative of the elderly, better grasp the disease status of the elderly, and improve the health literacy of the elderly through health information management.

The acceptability, affordability, and availability of community elderly care services have a significant impact on the psychological status of the elderly, which reflects that the resource attributes, service cost, and number of facilities of elderly care services also affect the psychological status of the elderly, indicating that the physical and psychology health of the elderly mainly depends on the accessibility of elderly care service resources. The more service resources for the elderly, the more collective social capital the elderly perceive, the capability of the elderly to participate in social activities becomes higher, and the psychological status of the elderly will be healthy. In addition, the accessible elderly care service resources will further stimulate the willingness of the elderly to participate in the utilization of elderly care services, increase the opportunities for physical exercise of the elderly, and improve the physical health of the elderly.

The accommodation of community elderly care services has a significant impact on the social relationships of the elderly, which reflects that the service process, service form, and organizational content of elderly care service can affect the formation of the social relations of the elderly. This is mainly due to the fact that different organization forms of elderly care services will affect the willingness of the elderly to participate in social activities. Service forms such as mutual support, group travel, and collective activities are conducive to increasing interaction and understanding among the elderly and promoting the formation of good social relations. In addition, the elderly can obtain social support and social roles in elderly care services and strengthen social relations among the elderly. Elderly care services in the form of door-to-door services or individual specialized services reduce the interaction opportunities of the elderly, which will make the elderly feel lonely and isolated.

The approachability, acceptability, accommodation, and availability of community elderly care services have a significant impact on living environment satisfaction, which reflects that the convenience, resource attribute, service process, and number of facilities of elderly care services will affect the living environment satisfaction of the elderly because the convenient and abundant construction of elderly care services can improve public transportation, activity places, service information, and public resources and improve the sense of fulfillment and happiness of the elderly.

In summary, the improvement of the accessibility of community elderly care services can enhance the capability of the elderly and directly improve the welfare of the elderly. This is similar to the research conclusion of Win et al. [30], who found that the improvement of the accessibility of healthcare services can improve the quality of life of migrant workers. The study of Saha et al. also found that the factors affecting the quality of life of Chinese farmers mainly included economic status, pension security, leisure and entertainment, social life, etc. [31]. These factors are highly consistent with the content of the accessibility of community elderly care services, which further proves the reliability of the conclusion of this article. Di et al. evaluated community elderly care services from the dimensions of accessibility and pointed out that community elderly care services could not meet the needs of the elderly in terms of facility resources, medical services, and nursing contents [32]. The result also proves that the quality of life of the elderly is highly dependent on the improvement of service accessibility, which is in line with the conclusions of this article.

Therefore, the government should improve the accessibility of community elderly care services. Firstly, improve the approachability of elderly care services by standardizing the construction of elderly care service facilities, increasing the supply of senior-friendly products, and accelerating the full coverage of elderly care service facilities. Secondly, improve the availability by establishing an elderly service platform, increasing elderly care service information publicity, expanding elderly care service publicity, and training elderly care service personnel. Thirdly, optimize the acceptability by improving the operability of the elderly care services, enriching the content of elderly care services, and improving the incentive mechanism of elderly services. Fourthly, improve the accommodation by increasing the precision of service objects and standardizing service construction and service data. Fifthly, improve the affordability by expanding financing channels for elderly care services, improving pricing mechanisms for elderly care services, and enhancing the economic ability of the elderly.

### 4.3. Indirect Impact

#### 4.3.1. Mediating Effect of Service Utilization Frequency

Taking the service utilization frequency as the mediator variable, we analyze the generation mechanism of the welfare effect of accessibility.

In Table 4, the approachability, acceptability, availability, and accommodation of community elderly care services have a significantly positive impact on the service utilization frequency and pass the significance test level of 5%, indicating that the improvement of the convenience of elderly care services, the number of facilities, the supply contents, and the service process can increase the service utilization frequency. However, affordability has no significant impact on the frequency of use, indicating that income has no significant impact on the frequency of service use. This is mainly due to the fact that the promotion and operation of community elderly care services in China mainly rely on government subsidies and community support, and most of the service contents are free or preferential, so economic ability is not the main reason that prevents the elderly from using elderly care services [33].

From the perspective of the impact of service utilization frequency on the welfare of the elderly, service utilization frequency only promotes the living environment satisfaction of the elderly, which indicates that the impact of accessibility on living environment satisfaction can be further influenced by influencing service utilization frequency. According to the results, the service utilization frequency has a complete mediating effect on accommodation and availability, indicating that the construction of elderly care service infrastructure and the optimization of the elderly care service process can improve the convenience of community life for the elderly. The living environment satisfaction of the elderly is mainly reflected in the process of participating in social activities. Therefore, the impact of accessibility on living environment satisfaction is completely reflected in the impact of service utilization frequency on living environment satisfaction. The service utilization frequency has a partial mediating effect on approachability and acceptability, indicating that the convenience of elderly care services, facility supply, and service resources have an impact on the elderly, which will not only enhance the capability of the elderly but also increase the utilization frequency of elderly care services [34]. However, the relationship between the improvement of physical condition, psychological status, social relationships of the elderly and the service utilization frequency has not yet emerged, maybe because the service utilization frequency only reflects the convenience of the elderly participating in elderly care services but cannot reflect the actual effect of participating in elderly care services.

#### 4.3.2. Mediating Effect of Service Utilization Satisfaction

Taking service utilization satisfaction as the mediator variable, we analyze the generation mechanism of the welfare effect of accessibility.

In Table 5, the approachability, accommodation, and availability of community elderly care services have a significant positive impact on service utilization satisfaction and pass the significance test level of 1%, indicating that the convenience, organizational form, and number of service facilities affect service utilization satisfaction. However, acceptability and affordability have no significant impact on service utilization satisfaction, indicating that the characteristics of elderly care service resources and the cost of elderly care service will not affect service utilization satisfaction. The elderly service facilities are mainly fitness equipment, community beds, and entertainment facilities, which are simple and easy to operate. The service contents mainly focus on community dining and leisure and entertainment, and the cost of elderly care services is low, so acceptability and affordability are not the main factors affecting service utilization satisfaction [35].

From the perspective of the impact of service utilization satisfaction on the welfare of the elderly, service utilization satisfaction only has a positive effect on the psychological status and living environment satisfaction of the elderly, indicating that the impact of accessibility on the psychological status and living environment satisfaction of the elderly can be further influenced by influencing the service utilization satisfaction. From the impact results of service utilization satisfaction on the psychological status of the elderly, service utilization satisfaction has a complete mediating effect on approachability, accommodation, and availability, indicating that the improvement of the psychological status of the elderly in the process of using elderly care service depends entirely on service utilization satisfaction [8]. From the impact results of service utilization satisfaction on the living environment satisfaction of the elderly, service utilization satisfaction has a partial mediating effect on approachability, accommodation, and availability, indicating that the living environment satisfaction depends to a certain extent on the service utilization satisfaction, mainly because convenient service places, sufficient fitness equipment, fast service process, etc., will improve the willingness of the elderly to use elderly care services and increase the elderly’s recognition of the living environment.

In summary, accessibility can improve the welfare of the elderly by influencing the utilization of the services. Acceptability can improve the quality of life of the elderly by affecting the frequency and satisfaction of service utilization, indicating that the features of the elderly care service facilities, service content, incentive mechanism and service quality can affect the satisfaction and sustainability of service utilization. Approachability, accommodation, and availability only have an impact on service utilization satisfaction, indicating that improvement of the elderly care service forms, distance, and number of resources can effectively meet the needs of the elderly [36]. However, the impact of affordability on service utilization is not significant, mainly because the current construction and development of elderly care services are still focusing on the construction of basic facilities and contents of elderly care services and have not provided effective support to the individual economy of the elderly.

So, we can improve the utilization efficiency by improving the quality of community elderly care services. In order to guide the elderly to participate in elderly care services, we must increase the supply of elderly products, encourage high-quality elderly products to enter communities, villages, and towns, and provide certain funds or policy subsidies to reduce the transportation costs and management costs of elderly products [37]. Secondly, community elderly care services should be evaluated regularly. The government should set up elderly care service evaluation groups, including government regulators, community service personnel, elderly targets, families, third-party organizations, etc., to regularly evaluate community elderly care services. In addition, we should establish a reward and punishment mechanism, forcing the quality of elderly care services to improve [38]. Finally, market service subjects should be encouraged to provide door-to-door service. With the endorsement of the government, market services can increase the elderly’s cognition and consumption willingness to elderly care services through door-to-door service and increase the enthusiasm of elderly people to participate in elderly care services.

## 5. Conclusions

From the perspective of accessibility, this paper discusses the influence mechanism of the accessibility of community elderly care services on the quality of life of the elderly. This research results show that, firstly, availability, acceptability, and affordability significantly affect the physical condition and psychological status, accommodation affects the social relationship, and approachability, availability, acceptability, and accommodation affect the living environment satisfaction of the elderly. Secondly, from the perspective of the intermediary path, the approachability, acceptability, availability, and accommodation of community elderly care services have a significantly positive impact on the service utilization frequency, and the approachability, accommodation, and availability have a significantly positive impact on service satisfaction, indicating that the improvement of accessibility can increase the willingness and behavior of services utilization and then improve the welfare of the elderly.

## 6. Limitations

In this paper, survey data in 2019 are used to evaluate the accessibility of elderly care services. The data are outdated, but they are still representative to a certain extent. In the future, we will conduct more investigations on the accessibility of elderly care services to analyze the changing characteristics and improvement paths of elderly care services. In addition, the measurement indicators of accessibility are subjective. With the increasing information on elderly care services, our next research will introduce macro measurement data to further improve the measurement system of accessibility.

## Figures and Tables

**Table 1 healthcare-13-00099-t001:** Descriptive statistics of sample.

Individual Characteristics	Frequency	Percentage	Individual Characteristics	Frequency	Percentage
Gender			Health status		
Male	260	39.39	Very bad	21	3.18
Female	400	60.61	Bad	112	16.97
Age			Average	182	27.58
60–65	136	20.61	Better	218	33.03
65–70	186	28.18	Well	127	19.24
70–75	155	23.48	Education		
75–80	98	14.85	Elementary	285	43.18
Above 80	85	12.88	Junior	187	28.33
Political status			High	133	20.15
Communist Party	196	29.70	Junior college	39	5.91
others	464	70.30	Bachelor	16	2.42

**Table 2 healthcare-13-00099-t002:** Variables, indicators, and sample characteristics.

Variable	Indicator	Question	Options	Mean	Median	Standard Deviation
The quality of life of the elderly	Physical Condition	How do you rate your physical condition	1. Very bad 2. Bad 3. Average 4. Better 5. Well	3.49	4	1.08
	Psychological status	How do you rate your psychology status	1. Very bad 2. Bad 3. Average 4. Better 5. Well	4.21	4	0.93
	Social relationship	Do you have a good relationship with your neighbors and friends	1. Very bad 2. Bad 3. Average 4. Better 5. Well	4.34	4	0.78
	Living environmental satisfaction	How do you rate your living environment of the community	1. Very bad 2. Bad 3. Average 4. Better 5. Well	3.61	4	1.33
Accessibility of community Elderly care services	Approachability	How do you rate convenience to get to the elderly care center from your home	1. Very bad 2. Bad 3. Average 4. Better 5. Well	3.78	4	1.43
	Availability	How do you rate degree of completeness of community elderly care services	1. Very bad 2. Bad 3. Average 4. Better 5. Well	3.05	3	1.12
	Acceptability	Do you trust elderly care workers	1. Hard to trust 2. Little to trust 3. Don’t care 4. Relatively trust 5. Very trust	3.62	4	1.27
	Accommodation	Would you like to use the community elderly care services	1. Don’t like very much 2. Don’t like 3. Don’t care 4. Like 5. Like very much	3.59	4	1.15
	Affordability	Total income for 2018	1. Less than 10 2. 10,000–30,000 3. 30,000–50,000 4. 50,000–100,000 5. more than 100,000	3.33	4	0.96
Mediator variable	Frequency of service utilization	How often do you use the community elderly care services	1. Never use 2. sometimes use 3. use 4. Often use 5. usually use	2.92	3	1.19
	Satisfaction of service utilization	How do you rate degree of satisfaction of community elderly care services	1. Very bad 2. Bad 3. Average 4. Better 5. Well	3.46	4	1.05
Control variables	Personal characteristics	Age	1. 60–65 2. 65–70 3. 70–75 4. 75–80 5. Above 80	3.16	3	1.08
		Gender	1. Male 0. Female	0.39	0	0.49
		Education	1. Elementary 2. Junior 3. High 4. Junior college 5. Bachelor	1.96	2	1.04
		Marriage	1. Yes 0. Other	0.71	1	0.45

**Table 3 healthcare-13-00099-t003:** Influence results of the accessibility on the welfare of elderly.

Variable	Physical Condition	Psychological Status	Social Relationship	Living Environmental Satisfaction
Approachability	0.0527 (0.3296)	0.0610 (0.2650)	−0.0129 (0.8197)	0.3099 *** (0.0000)
Acceptability	0.1491 ** (0.0125)	0.2278 *** (0.0003)	0.0947 (0.1354)	0.2164 *** (0.0002)
Affordability	0.2820 *** (0.0000)	0.1240 * (0.0758)	0.0470 (0.5105)	0.0033 (0.9610)
Accommodation	−0.0013 (0.9846)	0.0570 (0.4224)	0.2214 *** (0.0025)	0.2131 *** (0.0024)
Availability	0.1593 ** (0.0252)	0.2355 *** (0.0015)	0.0939 (0.2141)	0.2828 *** (0.0001)
Age	−0.2519 *** (0.0000)	0.0470 (0.4483)	−0.0677 (0.2745)	0.0435 (0.4521)
Gender	0.1919 (0.2170)	0.3832 ** (0.0202)	−0.1395 (0.3976)	0.0581 (0.7088)
Education	0.0312 (0.859)	0.2465 (0.1852)	−0.0982 (0.6083)	0.0075 (0.9670)
Marriage	0.0523 (0.5386)	0.0485 (0.5835)	0.0140 (0.8767)	0.1559 * (0.0629)
LR	70.67	52.76	24.73	113.88
Log likelihood	−906.17	−737.78	−667.18	−914.47

Note: *p*-values are in parentheses. *** means significant at the 1% level, ** means significant at the 5% level, and * means significant at the 10% level. LR is Chi-square statistics.

**Table 4 healthcare-13-00099-t004:** Influence results of mediating effect of service utilization frequency.

Variable	Service Utilization Frequency	Physical Condition	Psychology Status	Social Relationship	Living Environmental Satisfaction
Approachability	0.1307 ** (0.0204)	0.0574 (0.2893)	0.0576 (0.2947)	−0.0128 (0.8216)	0.2902 *** (0.0000)
Acceptability	0.2426 *** (0.0001)	0.1588 *** (0.0083)	0.2212 *** (0.0005)	0.0948 (0.1381)	0.1655 *** (0.0053)
Affordability	0.0787 (0.2533)	0.2873 *** (0.0000)	0.1204 * (0.0855)	0.0471 (0.5106)	−0.0144 (0.8310)
Accommodation	1.0489 *** (0.0000)	0.0505 (0.5245)	0.0263 (0.7461)	0.2221 *** (0.0081)	−0.0101 (0.8992)
Availability	0.7687 *** (0.0000)	0.1988 *** (0.0098)	0.2129 *** (0.0077)	0.0945 (0.2440)	0.1264 (0.1029)
Age	0.0297 (0.6191)	−0.2516 *** (0.0000)	0.0469 (0.4485)	−0.0677 (0.2748)	0.0371 (0.5229)
Gender	−0.0592 (0.7159)	0.1906 (0.220)	0.3850 ** (0.0197)	−0.1396 (0.3975)	0.0846 (0.5893)
Education	−0.2391 (0.2000)	0.0119 (0.9463)	0.2559 (0.1700)	−0.0984 (0.6081)	0.0717 (0.6948)
Marriage	0.0753 (0.3869)	0.0554 (0.5148)	0.0473 (0.5928)	0.0140 (0.8764)	0.1419 * (0.0952)
Service utilization frequency		−0.1123 (0.1740)	0.0667 (0.4413)	−0.0016 (0.9850)	0.4987 *** (0.0000)
LR	432.17	72.52	53.35	24.73	146.59
Log likelihood	−782.38	−905.24	−737.48	−667.18	−898.11

Note: *p*-values are in parentheses. *** means significant at the 1% level, ** means significant at the 5% level, and * means significant at the 10% level. LR is Chi-square statistics.

**Table 5 healthcare-13-00099-t005:** Influence results of mediating effect of service utilization satisfaction.

Variable	Service Utilization Satisfaction	Physical Condition	Psychology Status	Social Relationship	Living Environmental Satisfaction
Approachability	0.1515 *** (0.0075)	0.0408 (0.4533)	0.0430 (0.4368)	−0.0161 (0.7761)	0.2974 *** (0.0000)
Acceptability	0.1018 (0.1071)	0.1431 ** (0.0166)	0.2169 *** (0.0005)	0.0928 (0.1441)	0.2087 *** (0.0004)
Affordability	0.0084 (0.9063)	0.2790 *** (0.0000)	0.1227 * (0.0794)	0.0466 (0.5136)	0.0021 (0.9748)
Accommodation	0.3110 *** (0.0000)	−0.0132 (0.8495)	0.0345 (0.6293)	0.2164 *** (0.0034)	0.1969 *** (0.0053)
Availability	1.6871 *** (0.0000)	0.0662 (0.4688)	0.0909 (0.3348)	0.0597 (0.5383)	0.1811 * (0.0508)
Age	0.1094 * (0.0777)	−0.2577 *** (0.0000)	0.0392 (0.5281)	−0.0696 (0.2622)	0.0357 (0.5387)
Gender	−0.1185 (0.4790)	0.2015 (0.1955)	0.3946 ** (0.0169)	−0.1379 (0.4034)	0.0609 (0.6954)
Education	0.1240 (0.5225)	0.0280 (0.8736)	0.2344 (0.2069)	−0.1029 (0.5915)	−0.0170 (0.9256)
Marriage	−0.1382 (0.1214)	0.0646 (0.4496)	0.0615 (0.4879)	0.0169 (0.8511)	0.1636 * (0.0514)
Service utilization satisfaction		0.1596 (0.1063)	0.2522 ** (0.0133)	0.0582 (0.5735)	0.1757 * (0.0805)
LR	499.01	73.27	58.89	25.04	116.94
Log likelihood	−675.73	−904.87	−734.71	−667.02	−912.93

Note: *p*-values are in parentheses. *** means significant at the 1% level, ** means significant at the 5% level, and * means significant at the 10% level. LR is Chi-square statistics.

## Data Availability

Data are contained within the article.

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
