# Peer review of "The Impact of Accessibility of Community Elderly Care Services on Quality of Life of the Elderly"

_healthcare, 2025, doi:10.3390/healthcare13020099_

Round 1

Reviewer 1 Report

Comments and Suggestions for Authors
  1. Methodological Clarity:

    • While the study employs a robust sampling method and relevant variables, it would benefit from more detailed descriptions of how specific variables related to accessibility (e.g., approachability, availability, and affordability) were operationalized. This will help readers better understand the measurement process and strengthen the transparency of the methodology.
  2. Geographic Limitations:

    • The study focuses on Shaanxi province, which may limit the generalizability of the findings to other regions with different socio-economic or healthcare contexts. It would be beneficial to discuss how the findings might be applied to other areas of China or even internationally, particularly in regions with similar aging demographics.
  3. Integration of Technological Impact:

    • The role of technology in elderly care services (such as telemedicine or smart home technology) could be explored further, especially in urban areas where digital solutions are becoming more prevalent. This would add depth to your analysis and could potentially explain some of the barriers to service utilization in rural areas.
  4. Improvement of Service Descriptions:

    • In the discussion of service accessibility, consider expanding on the specific features of elderly care services in Shaanxi province. For example, what specific types of facilities or service models are currently being implemented, and how do they align with or differ from best practices in other regions?
  5. Further Explanation of Statistical Results:

    • The results section presents clear and significant findings, but it could benefit from a more thorough explanation of the implications of the statistical values (e.g., regression coefficients, p-values). A brief interpretation of how these results impact the elderly’s quality of life and decision-making could make the findings more actionable for policymakers and practitioners.
  6. Policy Implications:

    • The recommendations for improving elderly care services are valuable, but further elaboration on the role of local governments, community organizations, and private sectors in implementing these policies would strengthen the practical applicability of your conclusions.
  7. English Language and Presentation:

    • Overall, the English used in the article is clear, though a few minor grammatical improvements could make the presentation even more polished. For example, revising some sentences to avoid repetition or clarify certain points would enhance readability.

By addressing these areas, the paper can be further strengthened and provide even more practical insights for improving elderly care services in China and potentially other countries facing similar challenges.

Note

I have noticed that the plagiarism report indicates a 34% similarity, which may raise concerns about the originality of certain sections of your manuscript. It is important to ensure that all sources are properly cited, and any overlapping content is either reworded or adequately referenced. I recommend thoroughly reviewing your manuscript to verify that all borrowed ideas, data, and direct quotes are appropriately acknowledged. This will help to ensure compliance with academic integrity standards and the journal's guidelines.

If you are unsure about any sections, I suggest rephrasing or adding clearer citations to avoid any potential issues with plagiarism.

Author Response

Thank you very much for your suggestions and we made a point-to-point response. Details about the modification are in the revised version.

1.While the study employs a robust sampling method and relevant variables, it would benefit from more detailed descriptions of how specific variables related to accessibility (e.g., approachability, availability, and affordability) were operationalized.

Response: Approachability is measured by the question of how do you rate convenience to get to the elderly care center from your home, availability is measured by the question of how do you rate degree of completeness of community elderly care services,acceptability is measured by the question of do you trust elderly care workers, accommodation is measured by the question of would you like to use the community elderly care services, and affordability is measured by the question of total income for 2018. And we use the Likert method to evaluate five indicators, and the options are divided into five levels from weak to strong. The specific characteristics of the indicators are shown in Table 2.

2.The study focuses on Shaanxi province, which may limit the generalizability of the findings to other regions with different socio-economic or healthcare contexts. It would be beneficial to discuss how the findings might be applied to other areas of China or even internationally, particularly in regions with similar aging demographics.

Response: From the perspective of the elderly care service facilities, as of 2019, a total of 10,886 community service institutions and facilities had been built, with a total of 188,000 beds and 58,036 community day care beds in Shaanxi Province. There are 99,862 beds in elderly care institutions, 55,699 nursing staff in institutions, and 10,132 rural mutual support institutions. The number of elderly people receiving subsidies for elderly care services is 7,997. The construction of community elderly care services in Shaanxi is consistent with the development level of community elderly care services in most regions of China.Moreover, according to the data of the China Statistical Yearbook in 2023, the elderly dependency ratio in Shaanxi Province is 21.42, which is basically the same as the Chinese level of 21.83. Therefore, taking Shaanxi Province as an example to analyze the development status of China’s community elderly care service has a certain representative.

3.The role of technology in elderly care services (such as telemedicine or smart home technology) could be explored further, especially in urban areas where digital solutions are becoming more prevalent. This would add depth to your analysis and could potentially explain some of the barriers to service utilization in rural areas.

Response: We supplement the analysis in line 293. In addition, the application of smart technologies such as telemedicine or smart home technology can exert the subjective initiative of the elderly, better grasp the disease status of the elderly, and improve the health literacy of the elderly through health information management.

4.In the discussion of service accessibility, consider expanding on the specific features of elderly care services in Shaanxi province. For example, what specific types of facilities or service models are currently being implemented, and how do they align with or differ from best practices in other regions?

Response: In Shaanxi, the approachability of community elderly care services includes the construction of elderly care service facilities in residential communities, the establishment of family care beds, the provision of door-to-door care services, and the "15-minute" elderly care service circle. Availability includes the use of rural mutual support institutions and nursing homes, and the construction of information service platforms for elderly care. The acceptability includes the development of diversified insurance products, the establishment of long-term care insurance system, and the service quality of elderly care service personnel. The accommodation includes the establishment of a contract service mechanism for family doctors, and the provision of home medical services for the elderly with difficulties. Affordability includes subsidizing care for disabled seniors with financial difficulties and giving priority to elderly care services for those with financial difficulties.

These elderly care services are carried out in accordance with the Opinions on Promoting the Construction of Basic Elderly Care Service System issued by The State Council of China. Therefore, the study on the utilization of elderly care services in Shaanxi has certain reference significance for understanding the utilization of elderly care services in China.

5.The results section presents clear and significant findings, but it could benefit from a more thorough explanation of the implications of the statistical values (e.g., regression coefficients, p-values). A brief interpretation of how these results impact the elderly’s quality of life and decision-making could make the findings more actionable for policymakers and practitioners.

Response: We added the explanation of regression coefficient in line 284. That is, for every 1% increase in the approachability, affordability and availability of community elderly care services, the incidence rate of the elderly’s physical health improving by at least one level increased by 14.91%, 28.20% and 15.93%.

6.The recommendations for improving elderly care services are valuable, but further elaboration on the role of local governments, community organizations, and private sectors in implementing these policies would strengthen the practical applicability of your conclusions.

Response: In line 457, we explain the role of government in improving access to elderly care.In line 480, we explain the supervisory role of community service personnel and third-party agencies.

In the follow-up study, we will continue to discuss the role of different entities in improving elderly care services, and explain in the limited part.

7.Overall, the English used in the article is clear, though a few minor grammatical improvements could make the presentation even more polished. For example, revising some sentences to avoid repetition or clarify certain points would enhance readability.

Response: We have revised the sentences and grammar of the whole paper to avoid repetition.

8.I have noticed that the plagiarism report indicates a 34% similarity, which may raise concerns about the originality of certain sections of your manuscript. It is important to ensure that all sources are properly cited, and any overlapping content is either reworded or adequately referenced. I recommend thoroughly reviewing your manuscript to verify that all borrowed ideas, data, and direct quotes are appropriately acknowledged. This will help to ensure compliance with academic integrity standards and the journal's guidelines.

Response: The article with a higher similarity rate is my preprint. And we reviewed our manuscript and verified that all borrowed ideas, data, and direct quotes are appropriately acknowledged.

Reviewer 2 Report

Comments and Suggestions for Authors

Dear Author(s), 

Your manuscript is very interesting especially for the old persons and their accessibility to community elderly care services. However, find below some improvement suggestions for the manuscript:

- the abstract should be rephrased and information regarding the methodology, discussion and further research should also be provided. 

- in this study, 2019 data was used and I am wondering if this information is outdated. This fact should be specified in the limitations subsection, as described further below. 

- the analysis framework for the first hypothesis a bit too long. Please select only the essential information which would support this hypothesis. 

- in table 1, some information about the political status of the respondents has been included. I do not see the reason, maybe the authors may explain further the relevance of this information. 

- a statistical descriptive analysis should also be included in the methodology section. For instance, there is no specific information about the statistical software used and the statististical tests (the logistic regression, mean, median, standard deviation, etc) and the statistical significance threshold. 

- in the empirical analysis section a lot of information has been included, but I think that it is highly relevant to keep only the reported results without the discussion which would follow. It would be highly relevant to have a different section for the discussions. 

- I would also ask what is the relevance of the regression formula and other formula for the indirect impact. Have the authors used the formula and not a statistical software?

- was the mediation an expected result? If not, and it was not supported by any hypothesis, the authors should focus more on the unexpected results. 

- the discussion section should also contain other subsections: limitations and further research directions. The limitations section should contain all the information which can be a barrier for publishing the manuscript (having the database from 2019). The further research directions may be considered the suggestion part.

- the conclusion section should be very short and should emphasize the most important outcomes. 

Thank you!

Good luck!

Author Response

Thank you very much for your suggestions and we made a point-to-point response. Details about the modification are in the revised version.

1.the abstract should be rephrased and information regarding the methodology, discussion and further research should also be provided.

Response: We rephrase the abstract in line 20, and the methodology,  and add description of methods and results.

Improving the accessibility of community elderly care services is a basic requirement for coping with population ageing. Based on the survey data of elderly care services in Shaanxi Province in 2019, the paper uses logistic regression model to analyze the impact of accessibility on the quality of life of the elderly. The findings are as follows. Firstly, community elderly care services have significant effects on the physical health, psychological health, social relationship and environmental satisfaction of the elderly. Secondly, accessibility affects environmental satisfaction by influencing the frequency of service utilization. Thirdly, accessibility affects psychological health and environmental satisfaction of the elderly by influencing service utilization satisfaction. Finally, based on the empirical results, the paper puts forward some suggestions to improve the the elderly welfare from the high-quality development of elderly care services, access construction and efficient service utilization.

2.in this study, 2019 data was used and I am wondering if this information is outdated. This fact should be specified in the limitations subsection, as described further below.

Response: According to the data of the China Statistical Yearbook, in 2019, the elderly dependency ratio in Shaanxi Province Chinese mean value is 16.41and 17.8, in 2020, the elderly dependency ratio in Shaanxi Province Chinese mean value is 19.21 and 19.74, in 2021, the elderly dependency ratio in Shaanxi Province Chinese mean value is 20.33 and 20.82, in 2022, the elderly dependency ratio in Shaanxi Province Chinese mean value is 21.42 and 21.83. The status quo of elderly care services in Shaanxi Province is closest to the average level in China. Therefore, taking Shaanxi Province as an example to analyze the development status of China’s community elderly care service has a certain representative.

In addition, we explain the limitations of the fact that the data is out of date.

3.the analysis framework for the first hypothesis a bit too long. Please select only the essential information which would support this hypothesis.

Response: The paragraph preceding hypothesis 1 is an introduction to capability approach and the basis for the four hypotheses.

In addition, we deleted some unnecessary explanations for the hypothesis. For example, delete the sentence of “Therefore, Any barrier that restricts the capability of the elderly to participate in social activities will reduce the initiative, autonomy and selectivity of the elderly in performing functional activities”.

4.in table 1, some information about the political status of the respondents has been included. I do not see the reason, maybe the authors may explain further the relevance of this information.

Response:At present, the construction and development of elderly care services in China are more government-oriented services and supply, so members of the Communist Party of China may be more willing to use elderly care services. Therefore, in order to explain the rationality of the survey objects, this paper briefly lists the political status of the survey objects.

5.a statistical descriptive analysis should also be included in the methodology section. For instance, there is no specific information about the statistical software used and the statististical tests (the logistic regression, mean, median, standard deviation, etc) and the statistical significance threshold.

Response: we changed Section 3 to materials and methods and changed Section 4 to results and discussion. And add 3.3 Research Method. Statistical characteristics of all variables are shown in Table 2, and we add statistical descriptions of variables.

It can be seen from Table 2 that the standard deviation is close to 1, indicating that the data is concentrated near a certain central value, and the sample has no outliers. In terms of accessibility, the mean value and median value of availability are the lowest, 3.05 and 3 respectively, which shows that the infrastructure and products of community elderly care services are relatively lacking. From the perspective of the quality of life of the elderly, the mean value of environment satisfaction is general, indicating that the elderly have low satisfaction with resources such as community environment, public resources and interactive places. On the whole, the sample conforms to the actual situation of the older adults and has certain reliability.

3.3 Research Method

Firstly, In order to verify the direct welfare effect of the accessibility of community elderly care services, the paper respectively takes physical condition, psychology status, social relationship and living environmental satisfaction as dependent variables, and describes the impact of each dimension on the welfare of the elderly by constructing an Ordered Logistic Regression equation (Lall et al, 2002). The Ordered Logistic Regression is a Logit model based on cumulative distribution. Assuming that the dependent variable is an ordered value assigned from 1 to J, the cumulative Logit of the dependent variable  and  can be expressed as its basic theoretical model (1).

 (1)

In Formula (1),denotes dependent variables, specifically including core explanatory variables and control variables. denotes the matrix of coefficients corresponding to.denotes the sequence number assigned to the dependent variable from 1 to. denotes the intercept term.

According to the basic theoretical model, we can construct the following regression equation(2).

(2)

In the equation, denotes the dependent variable, that is the welfare of the elderly and its sub-dimensions, denotes the independent variable, that is the five dimensions of accessibility, denotes the control variable, denotes the regression coefficient, anddenotes the error term.

Then, the paper further explore the influence mechanism of the accessibility on the quality of life of the elderly, from the perspective of service utilization, we take the service utilization frequency and service utilization satisfaction as the mediator variable to analyze the welfare effect of the accessibility of community elderly care services. So, we can construct the following regression equation.

 (3)

 (4)

 (5)

In the equation, denotes service utilization, denotes accessibility, denotes control variable, 、、denotes regression coefficient, denotes error term.

Finally, the paper used stata15 software to analyze direct impact and indirect impact.

4.in the empirical analysis section a lot of information has been included, but I think that it is highly relevant to keep only the reported results without the discussion which would follow. It would be highly relevant to have a different section for the discussions.

Response: we add some discussions behind direct effect result.

Therefore, government should improve the accessibility of community elderly care services. Improve the approachability of elderly care services by standardizing the construction of elderly care service facilities, increasing the supply of senior-friendly products, and accelerating the full coverage of elderly care service facilities. Improve the availability by establishing the elderly service platform, increasing the elderly care service information publicity, expand elderly care service publicity and training elderly care service personnel. Optimize the acceptability by improving the operability of the elderly care services, enriching the content of the elderly care services and improving the incentive mechanism of the elderly service. Improve the accommodation by increasing precision of service objects, standardization of service construction and standardization of service data. Improve the affordability by expanding financing channels for elderly care services, improving pricing mechanisms for elderly care services, and enhancing the economic ability of the elderly.

we add some discussions behind indirect effect result.

So, we can improve the utilization efficiency by improving the quality of community elderly care services. in order to guide the elderly to participate in elderly care services, we must increase the supply of elderly products. Encourage high-quality elderly products to enter communities, villages and towns, and provide certain funds or policy subsidies to reduce the transportation costs and management costs of elderly products. Secondly, the community elderly care services should be evaluated regularly. The government should set up elderly care service evaluation groups, including government regulators, community service personnel, elderly targets, families, third-party organizations, etc., to regularly evaluate community elderly care services. Besides, we should establish a reward and punishment mechanism, forcing the quality of elderly care services to improve. Finally, market service subjects should be encouraged to provide door-to-door service. With the endorsement of the government, market services can increase the elderly's cognition and consumption willingness to elderly care services through door-to-door service, and increase the enthusiasm of elderly people to participate in elderly care services.

5.I would also ask what is the relevance of the regression formula and other formula for the indirect impact. Have the authors used the formula and not a statistical software?

Response: Firstly, based on logistics regression method, we built an analysis model through causal analysis to determine the dependent variables, independent variables, control variables and Mediator variables. Then, we used Stata 15 software to analyze the relationship between accessibility and quality of life of the elderly.

6.was the mediation an expected result? If not, and it was not supported by any hypothesis, the authors should focus more on the unexpected results.

Response: We have not proposed a hypothesis about the mediation, but have made a detailed analysis of the results of the mediation results. We add a lot of literature to discuss the results.

Chen Q, Amano T, Park S, et al. (2019) Home and community-based services and life satisfaction among homebound and poor older adults[J]. Journal of gerontological social work, 62(7): 708-727. DOI:10.1080/01634372.2019.1639094.

Dumka N, Ahmad T, Hannah E, et al. (2023). Health facility utilization and healthcare-seeking behaviour of the elderly population in India[J]. Journal of Family Medicine and Primary Care, 12(5): 902-916. DOI:10.4103/jfmpc.jfmpc_553_22.

Huang G, Guo F, Chen G.(2023) Utilization of home-/community-based care services: The current experience and the intention for future utilization in urban China[J]. Population Research and Policy Review, 42(4): 61. DOI:10.1007/s11113-023-09810-1.

Liu T Y, Qiu D C, Chen T. (2022). Effects of social participation by middle-aged and elderly residents on the utilization of medical services: evidence from China[J]. Frontiers in Public Health, 10: 824514. DOI:10.3389/fpubh.2022.824514.

Mozhaeva I.(2022). Inequalities in utilization of institutional care among older people in Estonia[J]. Health Policy, 126(7): 704-714. DOI:10.1016/j.healthpol.2022.04.008.

Samanta R, Munda J, Mandal S, et al. (2023). Health-care utilisation among India’s middle and older aged migrants: scrutinizing the status and predictors using Andersen’s simplified healthcare utilisation framework[J]. International Journal of Migration, Health and Social Care, 19(2): 142-156.DOI:10.1108/IJMHSC-07-2022-0068.

Yang L, Wang L, Dai X. (2021). Rural-urban and gender differences in the association between community care services and elderly individuals’ mental health: a case from Shaanxi Province, China[J]. BMC health services research, 21, 106. DOI:10.1186/s12913-021-06113-z.

Zhang H, Zhang S. (2024). Reasons for underutilization of community care facilities for the elderly in China[J]. BMC geriatrics, 24(1): 791. DOI:10.1186/s12877-024-05398-z.

7.the discussion section should also contain other subsections: limitations and further research directions. The limitations section should contain all the information which can be a barrier for publishing the manuscript (having the database from 2019). The further research directions may be considered the suggestion part.

Response: In this paper, survey data in 2019 are used to evaluate the accessibility of elderly care services. The data is outdated, but it is still representative to a certain extent. In the future, we will conduct more investigation on the accessibility of elderly care services to analyze the changing characteristics and improvement paths of elderly care services. In addition, the measurement indicators of accessibility is subjective. With the increasing information of elderly care services, our next research will introduce macro measurement data to further improve the measurement system of accessibility.

8.the conclusion section should be very short and should emphasize the most important outcomes.

Response: We deleted the suggestions section and moved the relevant content to the results and discussion section. In addition, we added limitations section.

Reviewer 3 Report

Comments and Suggestions for Authors

Dear Editor,

Thank you for inviting me to review the manuscript. This manuscript reports a study regarding the impact of community elderly care service. While the topic is interesting, there are several unclear parts in the manuscript that need to be clarified. The detailed comments can be found below.

  1. What is the aim of this paper? Please add.
  2. Hypothesis 4 does not have any explanation. Please provide the elaboration of hypothesis 4 as the authors did for the other three hypotheses.
  3. Section 3. should be changed to “materials and methods,” not research design.
  4. What is the design of this study? Please add another section for research design under section 3 (the materials and method) and explain what the design of this study is.
  5. Please add another section for data analysis under section 3 (materials and methods).
  6. Please change section 4 to results and move tables 1 and 2 from section 3 to section 4. Tables 1 and 2 should belong to the results section, not the research design section.
  7. How many respondents participated in this study? On page 5, line 226, it is written that, “289 pieces of survey data with obviously missing data were excluded.” However, in Table 1, the total of male and female respondents counts for 660. Please clarify.
  8. For the questionnaire, how did the authors decide the questions and the scores? Was it a self-made questionnaire, or derived from an existing questionnaire? Please explain.
  9. For formulas on pages 7, 8, and 9, are they author-made formulas or existing formulas? If existing formulas, please provide the reference. If it is author-made formulas, please explain how the authors formulate those formulas.
  10. Pages 7 and 9, tables 3 and 4, Please add the legend for Table 3. What does LR stand for? What do ** and *** mean, and how do they differ from each other?
  11. Pages 7 and 9, tables 3 and 4: What do the values outside of brackets and inside of brackets refer to? Are they mean, median, or p-value? Please clarify.
  12. Page 6, Table 2, please give the line to separate which indicators belong to each variable. In this version, it is not clear which indicator belongs to the quality of life, which indicator belongs to the accessibility, and so on.
  13. Overall:
  • Please remove the repetitive words: page 1 line 27, “some suggestions to improve the the elderly.” Please remove one “the.”.
  • Please use the consistent fonts and font size for the entire manuscript following the journal guidelines. In this current version, some parts of the manuscript use different fonts and sizes.
  • Academic writing needs to be revised. For example: Please use a capital letter for the beginning of a sentence. Page 2, line 77, the sentence begins with “accessibility”; it does not start with the capital letter. Please revise.
  • Professional proofreading for English is necessary; some sentences are not grammatically correct, for example, page 2, line 79, “the user can accessibility,” This sentence needs to be revised.
Comments on the Quality of English Language

Professional proofreading is necessary.

Author Response

Thank you very much for your suggestions and we made a point-to-point response. Details about the modification are in the revised version.

1.What is the aim of this paper? Please add.

Response: we add the aim of this paper in line 66.

In order to elucidate the influence mechanism of the accessibility of community elderly care services on the quality of life of the elderly.

2.Hypothesis 4 does not have any explanation. Please provide the elaboration of hypothesis 4 as the authors did for the other three hypotheses.

Response: Hypothesis 4 is explained in the preceding paragraph.

From the perspective of environmental adaptation ability of the elderly, the increase of public facilities for elderly care services, the elderly-oriented environmental reform and the enrichment of community elderly care service contents can greatly improve the care facility and community environment. A good community environment can effectively increase the social participation and neighborhood interaction of the elderly. Convenient transportation and sufficient elderly care resources can accelerate the circulation of elderly care service information, enrich the supply of elderly care service contents, and increase the selection range of elderly care service models, then improve the freedom of the elderly. Efficient service processes and high-quality elderly care services can improve the trust and dependence of the elderly on community services, and increase the cohesion between residents and community, as well as between residents (Nam & Kim, 2021). Therefore, the improvement of the accessibility of elderly care services can release the family pressure, replace the family care function with community care, and improve the life satisfaction of the elderly. Kiik & Nuwa (2020), based on cross-sectional study, found that the level of community elderly care service facilities could significantly improve the subjective welfare of the elderly, and the welfare effects of different types of community elderly care service facilities were different. Gu et al (2021) found that community elderly care services can effectively improve the elderly’s environmental satisfaction, and the more service items, the higher the elderly’s satisfaction. So, we can come up with hypothesis4 that the accessibility of community elderly care services can improve the environmental satisfaction of the elderly.

3.Section 3. should be changed to “materials and methods,” not research design. What is the design of this study? Please add another section for research design under section 3 (the materials and method) and explain what the design of this study is. Please add another section for data analysis under section 3 (materials and methods). Please change section 4 to results and move tables 1 and 2 from section 3 to section 4. Tables 1 and 2 should belong to the results section, not the research design section.

Response: we changed Section 3 to materials and methods and changed Section 4 to results. And add 3.3 Research Method.

3.3 Research Method

Firstly, In order to verify the direct welfare effect of the accessibility of community elderly care services, the paper respectively takes physical condition, psychology status, social relationship and living environmental satisfaction as dependent variables, and describes the impact of each dimension on the welfare of the elderly by constructing an Ordered Logistic Regression equation (Lall et al, 2002). The Ordered Logistic Regression is a Logit model based on cumulative distribution. Assuming that the dependent variable is an ordered value assigned from 1 to J, the cumulative Logit of the dependent variable  and  can be expressed as its basic theoretical model (1).

 (1)

In Formula (1),denotes explanatory variables, specifically including core explanatory variables and control variables. denotes the matrix of coefficients corresponding to.denotes the sequence number assigned to the dependent variable from 1 to. denotes the intercept term.

According to the basic theoretical model, we can construct the following regression equation(2).

(2)

In the equation, denotes the dependent variable, that is the welfare of the elderly and its sub-dimensions, denotes the independent variable, that is the five dimensions of accessibility, denotes the control variable, denotes the regression coefficient, anddenotes the error term.

Then, the paper further explore the influence mechanism of the accessibility on the quality of life of the elderly, from the perspective of service utilization, we take the service utilization frequency and service utilization satisfaction as the mediator variable to analyze the welfare effect of the accessibility of community elderly care services. So, we can construct the following regression equation.

 (3)

 (4)

 (5)

In the equation, denotes service utilization, denotes accessibility, denotes control variable, 、、denotes regression coefficient, denotes error term.

Finally, the paper used stata15 software to analyze direct impact and indirect impact.

4.How many respondents participated in this study? On page 5, line 226, it is written that, “289 pieces of survey data with obviously missing data were excluded.” However, in Table 1, the total of male and female respondents counts for 660. Please clarify.

Response: line 231, We made an explanation about the number of questionnaires.

A total of 949 pieces of survey data were collected, and 289 pieces of survey data with obviously missing data were excluded. So, the sample size is 660.

5.For the questionnaire, how did the authors decide the questions and the scores? Was it a self-made questionnaire, or derived from an existing questionnaire? Please explain.

Response: We used self-made questionnaire. In Table 2, we listed the measurement questions and options for each indicator, and use the Likert method to evaluate indicators, and the options are divided into five levels from weak to strong.

6.For formulas on pages 7, 8, and 9, are they author-made formulas or existing formulas? If existing formulas, please provide the reference. If it is author-made formulas, please explain how the authors formulate those formulas.

Response: We add the basic theoretical model of Ordered Logistic Regression (Lall et al, 2002).

Firstly, In order to verify the direct welfare effect of the accessibility of community elderly care services, the paper respectively takes physical condition, psychology status, social relationship and living environmental satisfaction as dependent variables, and describes the impact of each dimension on the welfare of the elderly by constructing an Ordered Logistic Regression equation (Lall et al, 2002). The Ordered Logistic Regression is a Logit model based on cumulative distribution. Assuming that the dependent variable is an ordered value assigned from 1 to J, the cumulative Logit of the dependent variable  and  can be expressed as its basic theoretical model (1).

 (1)

In Formula (1),denotes explanatory variables, specifically including core explanatory variables and control variables. denotes the matrix of coefficients corresponding to.denotes the sequence number assigned to the dependent variable from 1 to. denotes the intercept term.

According to the basic theoretical model, we can construct the following regression equation(2).

(2)

In the equation, denotes the dependent variable, that is the welfare of the elderly and its sub-dimensions, denotes the independent variable, that is the five dimensions of accessibility, denotes the control variable, denotes the regression coefficient, anddenotes the error term.

Then, the paper further explore the influence mechanism of the accessibility on the quality of life of the elderly, from the perspective of service utilization, we take the service utilization frequency and service utilization satisfaction as the mediator variable to analyze the welfare effect of the accessibility of community elderly care services. So, we can construct the following regression equation.

 (3)

 (4)

 (5)

In the equation, denotes service utilization, denotes accessibility, denotes control variable, 、、denotes regression coefficient, denotes error term.

7.Pages 7 and 9, tables 3 and 4, Please add the legend for Table 3. What does LR stand for? What do ** and *** mean, and how do they differ from each other? What do the values outside of brackets and inside of brackets refer to? Are they mean, median, or p-value? Please clarify. 

Response: We add the note in table 3 that P-values are in parentheses. *** means significant at the 1% level, ** means significant at the 5% level, and * means significant at the 10% level. LR is Chi-square statistics.

8.Page 6, Table 2, please give the line to separate which indicators belong to each variable. In this version, it is not clear which indicator belongs to the quality of life, which indicator belongs to the accessibility, and so on.

Response: We have modified it as required.

9.Please remove the repetitive words: page 1 line 27, “some suggestions to improve the the elderly.” Please remove one “the.”.

Response: We have modified it as required.

10.Please use the consistent fonts and font size for the entire manuscript following the journal guidelines. In this current version, some parts of the manuscript use different fonts and sizes.

Response: We have modified it as required.

11.Academic writing needs to be revised. For example: Please use a capital letter for the beginning of a sentence. Page 2, line 77, the sentence begins with “accessibility”; it does not start with the capital letter. Please revise.

Response: We have modified it as required.

  1. Professional proofreading for English is necessary; some sentences are not grammatically correct, for example, page 2, line 79, “the user can accessibility,” This sentence needs to be revised.

Response: We have modified it as required.

Reviewer 4 Report

Comments and Suggestions for Authors

The abstract does not reflect the paper, the abstract needs to be better structured, describing the methods, main results and conclusions.

In the introductory part of the paper, it is necessary to refer to literature sources in the part when you state data on the number of elderly people in China.

In the methods, it is necessary to describe how the data were collected. How did the respondents answer the questions? Is it necessary to describe how the questionnaire used in the research was created? Were the questions taken from another questionnaire or were they created for the purpose of this research? Are the conclusions about a particular dimension of quality of life presented on the basis of one question per field and results on a Likert scale? If they are, it should be stated in the limitations of the research.

When interpreting the direct and indirect impact, it is necessary to connect your results with the results of a larger number of published studies.

State the limitations of your paper. 

Comments on the Quality of English Language

The paper needs language editing.

Author Response

Thank you very much for your suggestions and we made a point-to-point response. Details about the modification are in the revised version.

1.The abstract does not reflect the paper, the abstract needs to be better structured, describing the methods, main results and conclusions.

Response: We have revised the abstract.

Improving the accessibility of community elderly care services is a basic requirement for coping with population ageing. Based on the survey data of elderly care services in Shaanxi Province in 2019, the paper uses logistic regression model to analyze the impact of accessibility on the quality of life of the elderly. The findings are as follows. Firstly, community elderly care services have significant effects on the physical health, psychological health, social relationship and environmental satisfaction of the elderly. Secondly, accessibility affects environmental satisfaction by influencing the frequency of service utilization frequency. Thirdly, accessibility affects psychological health and environmental satisfaction of the elderly by influencing service utilization satisfaction, indicating that the improvement of accessibility can increase the willingness and behavior of the services utilization, and then improve the welfare of the elderly.

2.In the introductory part of the paper, it is necessary to refer to literature sources in the part when you state data on the number of elderly people in China.

Response: We add a literature to the introductory part .

Wu, W., Long, S., Cerda, A.A. et al. (2023). Population ageing and sustainability of healthcare financing in China. Cost Eff Resour Alloc, 21, 97 . DOI:10.1186/s12962-023-00505-0.

3.In the methods, it is necessary to describe how the data were collected. How did the respondents answer the questions? Is it necessary to describe how the questionnaire used in the research was created? Were the questions taken from another questionnaire or were they created for the purpose of this research? Are the conclusions about a particular dimension of quality of life presented on the basis of one question per field and results on a Likert scale? If they are, it should be stated in the limitations of the research.

Response: The questionnaire is specially designed according to the content of the accessibility of elderly care services. In the process of survey, we adopted the method of random interview. Each interview group had two members, one responsible for interviewing the elderly, and the other responsible for filling in the questionnaire. After the interview, Interview group cross-checked the filling results of the questionnaire to ensure the accuracy of the questionnaire.

We add the limitation part. The measurement indicators of accessibility is subjective. With the increasing information of elderly care services, our next research will introduce macro measurement data to further improve the measurement system of accessibility.

4.When interpreting the direct and indirect impact, it is necessary to connect your results with the results of a larger number of published studies.

Response: We add a lot of literature to the discussion section.

Chen Q, Amano T, Park S, et al. (2019) Home and community-based services and life satisfaction among homebound and poor older adults[J]. Journal of gerontological social work, 62(7): 708-727. DOI:10.1080/01634372.2019.1639094.

Dumka N, Ahmad T, Hannah E, et al. (2023). Health facility utilization and healthcare-seeking behaviour of the elderly population in India[J]. Journal of Family Medicine and Primary Care, 12(5): 902-916. DOI:10.4103/jfmpc.jfmpc_553_22.

Huang G, Guo F, Chen G.(2023) Utilization of home-/community-based care services: The current experience and the intention for future utilization in urban China[J]. Population Research and Policy Review, 42(4): 61. DOI:10.1007/s11113-023-09810-1.

Liu T Y, Qiu D C, Chen T. (2022). Effects of social participation by middle-aged and elderly residents on the utilization of medical services: evidence from China[J]. Frontiers in Public Health, 10: 824514. DOI:10.3389/fpubh.2022.824514.

Mozhaeva I.(2022). Inequalities in utilization of institutional care among older people in Estonia[J]. Health Policy, 126(7): 704-714. DOI:10.1016/j.healthpol.2022.04.008.

Samanta R, Munda J, Mandal S, et al. (2023). Health-care utilisation among India’s middle and older aged migrants: scrutinizing the status and predictors using Andersen’s simplified healthcare utilisation framework[J]. International Journal of Migration, Health and Social Care, 19(2): 142-156.DOI:10.1108/IJMHSC-07-2022-0068.

Yang L, Wang L, Dai X. (2021). Rural-urban and gender differences in the association between community care services and elderly individuals’ mental health: a case from Shaanxi Province, China[J]. BMC health services research, 21, 106. DOI:10.1186/s12913-021-06113-z.

Zhang H, Zhang S. (2024). Reasons for underutilization of community care facilities for the elderly in China[J]. BMC geriatrics, 24(1): 791. DOI:10.1186/s12877-024-05398-z.

5.State the limitations of your paper.

Response: we add the limitations.

In this paper, survey data in 2019 are used to evaluate the accessibility of elderly care services. The data is outdated, but it is still representative to a certain extent. In the future, we will conduct more investigation on the accessibility of elderly care services to analyze the changing characteristics and improvement paths of elderly care services. In addition, the measurement indicators of accessibility is subjective. With the increasing information of elderly care services, our next research will introduce macro measurement data to further improve the measurement system of accessibility.

Round 2

Reviewer 2 Report

Comments and Suggestions for Authors

No further comments. All suggestions have been fully addressed. 

Author Response

Thank you very much for your suggestions again.

Reviewer 3 Report

Comments and Suggestions for Authors

Dear Editor,

Thank you for inviting me to review the revised version of the manuscript. I appreciate the authors’ efforts to revise the manuscript. Some minor comments of the previous review have been addressed such as adding the objectives of the study, removing hypothesis 4, changing the name of heading 3 to “materials and methods” section. Unfortunately, other previous comments have not been addressed well, including moving Tables 1 and 2 to the results section, clarification of the numbers of participants, etc. Moreover, the authors uploaded the file with tracking changes with showing all marks that made the file difficult to read. These track changes with all marks made it unclear which parts have been finally revised. The authors also did not provide responses to reviewers, which makes it not clear which comments have been addressed by them.

I suggest the authors prepare responses to reviewers in table form and make a clear response for each comment and indicate clearly where the revisions can be found in the manuscript file. Also, please make sure to upload a clean file without track changes and highlight the revised parts with a different color. By doing so, the revised manuscript might be read and reviewed easily. Thank you very much.

Comments on the Quality of English Language

The English could be improved to more clearly express the research.

Author Response

Thank you very much for your suggestions and we made a point-to-point response in table form. Details about the modification are in the revised version.

Number

Suggestions

Response

1

What is the aim of this paper? Please add.

We add the aim of this paper in line 66.

In order to elucidate the influence mechanism of the accessibility of community elderly care services on the quality of life of the elderly.

2

Hypothesis 4 does not have any explanation. Please provide the elaboration of hypothesis 4 as the authors did for the other three hypotheses.

Hypothesis 4 is explained in the preceding paragraph in line 188 to 209.

From the perspective of environmental adaptation ability of the elderly, the increase of public facilities for elderly care services, the elderly-oriented environmental reform and the enrichment of community elderly care service contents can greatly improve the care facility and community environment. A good community environment can effectively increase the social participation and neighborhood interaction of the elderly. Convenient transportation and sufficient elderly care resources can accelerate the circulation of elderly care service information, enrich the supply of elderly care service contents, and increase the selection range of elderly care service models, then improve the freedom of the elderly. Efficient service processes and high-quality elderly care services can improve the trust and dependence of the elderly on community services, and increase the cohesion between residents and community, as well as between residents (Nam & Kim, 2021). Therefore, the improvement of the accessibility of elderly care services can release the family pressure, replace the family care function with community care, and improve the life satisfaction of the elderly. Kiik & Nuwa (2020), based on cross-sectional study, found that the level of community elderly care service facilities could significantly improve the subjective welfare of the elderly, and the welfare effects of different types of community elderly care service facilities were different. Gu et al (2021) found that community elderly care services can effectively improve the elderly’s environmental satisfaction, and the more service items, the higher the elderly’s satisfaction. So, we can come up with hypothesis4 that the accessibility of community elderly care services can improve the environmental satisfaction of the elderly.

3

Section 3. should be changed to “materials and methods,” not research design. What is the design of this study? Please add another section for research design under section 3 (the materials and method) and explain what the design of this study is. Please add another section for data analysis under section 3 (materials and methods). 

We changed Section 3 to materials and methods and changed Section 4 to results and discussions. And add 3.3 Research Method in line 258-292.

3.3 Research Method

Firstly, In order to verify the direct welfare effect of the accessibility of community elderly care services, the paper respectively takes physical condition, psychology status, social relationship and living environmental satisfaction as dependent variables, and describes the impact of each dimension on the welfare of the elderly by constructing an Ordered Logistic Regression equation (Lall et al, 2002). The Ordered Logistic Regression is a Logit model based on cumulative distribution. Assuming that the dependent variable is an ordered value assigned from 1 to J, the cumulative Logit of the dependent variable  and  can be expressed as its basic theoretical model (1).

In Formula (1),denotes explanatory variables, specifically including core explanatory variables and control variables. denotes the matrix of coefficients corresponding to.denotes the sequence number assigned to the dependent variable from 1 to. denotes the intercept term.

According to the basic theoretical model, we can construct the following regression equation(2).

(2)

In the equation, denotes the dependent variable, that is the welfare of the elderly and its sub-dimensions, denotes the independent variable, that is the five dimensions of accessibility, denotes the control variable, denotes the regression coefficient, anddenotes the error term.

Then, the paper further explore the influence mechanism of the accessibility on the quality of life of the elderly, from the perspective of service utilization, we take the service utilization frequency and service utilization satisfaction as the mediator variable to analyze the welfare effect of the accessibility of community elderly care services. So, we can construct the following regression equation.

 (3)

 (4)

 (5)

In the equation, denotes service utilization, denotes accessibility, denotes control variable, 、、denotes regression coefficient, denotes error term.

Finally, the paper used stata15 software to analyze direct impact and indirect impact.

4

How many respondents participated in this study? On page 5, line 226, it is written that, “289 pieces of survey data with obviously missing data were excluded.” However, in Table 1, the total of male and female respondents counts for 660. Please clarify.

The survey team consists of 25 people, including teachers, doctoral students and master’s students in line 213.

We surveyed a total of 949 elderly people and adopted interview questionnaire survey method, so a total of 949 questionnaires were collected in line 295.

A total of 949 pieces of survey data were collected, and 289 pieces of survey data with obviously missing data were excluded. So, the sample size is 660.

5

For the questionnaire, how did the authors decide the questions and the scores? Was it a self-made questionnaire, or derived from an existing questionnaire? Please explain.

We used self-made questionnaire. In Table 2, we listed the measurement questions and options for each indicator, and use the Likert method to evaluate indicators, and the options are divided into five levels from weak to strong.

Most of the existing accessibility questionnaires are health care accessibility, and the framework is derived from the five-A model of Penchansky R and Thomas J (1981), i.e approachability, availability, acceptability, accommodation and affordability. So, based on the five-A model of Penchansky R and Thomas J, and combined with the characteristics of elderly care services, we built a measurement system and questionnaire suitable for the accessibility of elderly care services.

6

For formulas on pages 7, 8, and 9, are they author-made formulas or existing formulas? If existing formulas, please provide the reference. If it is author-made formulas, please explain how the authors formulate those formulas.

We add the basic theoretical model of Ordered Logistic Regression (Lall et al, 2002) in line 259-292.

7

Pages 7 and 9, tables 3 and 4, Please add the legend for Table 3. What does LR stand for? What do ** and *** mean, and how do they differ from each other? What do the values outside of brackets and inside of brackets refer to? Are they mean, median, or p-value? Please clarify. 

We add the note in table 3 that P-values are in parentheses. *** means significant at the 1% level, ** means significant at the 5% level, and * means significant at the 10% level. LR is Chi-square statistics in line 318-320.

8

Page 6, Table 2, please give the line to separate which indicators belong to each variable. In this version, it is not clear which indicator belongs to the quality of life, which indicator belongs to the accessibility, and so on.

We have modified it as required.

9

Please change section 4 to results and move tables 1 and 2 from section 3 to section 4. Tables 1 and 2 should belong to the results section, not the research design section.

We moved Tables 1 and 2 to the results section as 4.1 Characteristics of the sample in line 294-315.

10

Please remove the repetitive words: page 1 line 27, “some suggestions to improve the the elderly.” Please remove one “the.”.

 We have modified it as required.

11

Please use the consistent fonts and font size for the entire manuscript following the journal guidelines. In this current version, some parts of the manuscript use different fonts and sizes.

We have modified it as required.

12

Academic writing needs to be revised. For example: Please use a capital letter for the beginning of a sentence. Page 2, line 77, the sentence begins with “accessibility”; it does not start with the capital letter. Please revise.

We have modified it as required.

13

Professional proofreading for English is necessary; some sentences are not grammatically correct, for example, page 2, line 79, “the user can accessibility,” This sentence needs to be revised.

We have modified it as required.

Reviewer 4 Report

Comments and Suggestions for Authors

Dear authors,

Thank you for the revisions of the paper. There is a need for additional revision. The limitations should be placed at the end of the discussion, not as a part of the conclusions.

Author Response

Thank you very much for your suggestions again.

Response: We have modified it as required in line 511-519.

  1. Limitations

In this paper, survey data in 2019 are used to evaluate the accessibility of elderly care services. The data is outdated, but it is still representative to a certain extent. In the future, we will conduct more investigation on the accessibility of elderly care services to analyze the changing characteristics and improvement paths of elderly care services. In addition, the measurement indicators of accessibility is subjective. With the increasing information of elderly care services, our next research will introduce macro measurement data to further improve the measurement system of accessibility.
